# Employing Spent Frying Oil as a Feedstock to Produce Short-Chain Organic Acids Using Mixed Microbial Cultures

André Oliveira [1], Sílvia Petronilho [2,3,*] and Luísa S. Serafim [1,*]

1    CICECO-Aveiro Institute of Materials, Chemistry Department, Campus Universitário de Santiago, University of Aveiro, 3810-193 Aveiro, Portugal; andreoliveira98@ua.pt
2    LAQV-REQUIMTE, Department of Chemistry, Campus Universitário de Santiago, University of Aveiro, 3810-193 Aveiro, Portugal
3    Chemistry Research Centre-Vila Real, Department of Chemistry, University of Trás-os-Montes and Alto Douro, Quinta de Prados, 5001-801 Vila Real, Portugal
*    Correspondence: silviapetronilho@ua.pt (S.P.); luisa.serafim@ua.pt (L.S.S.)

**Abstract:** Food industry waste and wastewater have been explored in relation to acidogenic fermentation as sources of non-competing food carbohydrates and mixed microbial cultures (MMCs), respectively, with the aim of producing short-chain organic acids (SCOAs) with general applications in polyhydroxyalkanoates (PHAs) production. However, studies on acidogenic fermentation using lipidic substrates are scarce. In this work, it was hypothesized that spent frying oil (SFO) could be used as a substrate for SCOA production via MMCs. In this study, oleic acid was used as a model molecule. The characterization of SFO revealed that it is mainly composed of oleic acid (81%), with minor amounts of palmitic, linoleic, and stearic acids. Different MMCs and food-to-microorganism (F/M) ratios were tested. MMCs collected in the aerobic tank of a municipal wastewater treatment plant (AES), at a 1:1 F/M, allowed to obtain the highest SCOA concentration (1.50 g COD/L) and the most diverse profile of SCOAs, with the production of acetic, propionic, butyric, iso-butyric, and valeric acids at 48:17:9:13:13% on a molar basis, respectively. This variety of odd and even SCOAs is of upmost importance, with potential applications in producing PHAs. This work can be considered a starting point for future acidogenic fermentation studies using lipid-based substrates and for the future production of PHAs.

**Keywords:** short-chain organic acids; acidogenic fermentation; mixed microbial cultures; spent frying oil; oleic acid; waste valorization





## 1. Introduction

According to the European Environment Agency, 50.3 M tons of urban waste is produced yearly in EU, mainly comprising sewage sludge (28.0%), household and food industry waste (23.0%), and manufacturing waste (21.0%) [1,2]. Due to the high waste volumes produced yearly, circular economy and sustainable concepts have started being adopted, envisaging environmental sustainability [3,4].

To reduce waste volume regarding wastewater and the food industry while avoiding environmental pollution, their conversion into added value products has been explored [5]. These include short-chain organic acids (SCOAs), i.e., carboxylic acids composed of 2 to 8 carbon atoms, such as acetic, propionic, and butyric acids, among others [6]. SCOAs are commonly applied in the manufacture of other chemicals, pharmaceuticals, food, and agricultural products [7], including polyhydroxyalkanoates (PHAs), chemicals (such as esters, ketones, and alkanes), and biofuels (such as biogas and hydrogen) [3,8]. SCOAs are traditionally produced through chemical synthesis [9,10], commonly using fossil feedstocks, despite the associated economic and ecological concerns [10,11]. SCOAs are also intermediates of anaerobic digestion [12], which is a process that firstly involves the hydrolysis of proteins, carbohydrates, and lipids of a substrate into simple molecules, such

as amino acids, sugars, and fatty acids, respectively. These simple molecules are then converted into SCOAs, such as acetic, propionic, butyric, iso-butyric, and valeric acids, in acidogenesis and acetogenic steps. In the last step, known as methanogenesis, SCOAs are converted into $CO_2$ and methane. To maximize SCOA yields, methanogenesis must be inhibited [13,14], with previous studies using either pH control or specific or non-specific inhibitors [15–17]. SCOA production through anaerobic digestion is performed using mixed microbial cultures (MMCs), which are populations with an undefined composition that result from the transient characteristics of the process. MMCs can adapt to a wide range of substrates, such as agricultural, food, and industrial wastes, while offering energy and cost savings since the corresponding operations occur under non-sterile conditions [18,19]. SCOAs produced using this strategy are particularly interesting for the biosynthesis of polyhydroxyalkanoates (PHAs) since there is no need to separate the mixtures of SCOAs formed. For this reason, the most utilized process for the conversion of complex waste to PHAs via MMCs is the three-step process, in which the acidogenic fermentation of the waste is the first step [20]. Moreover, by manipulating the compositions of the mixtures obtained, it is possible to tailor the molecular compositions of PHAs and, consequently, their thermochemical characteristics [21].

Over the years, many researchers have tried, via anaerobic digestion, to produce SCOAs (through acidogenic fermentation) with complex waste or industry byproducts. These byproducts include tuna waste [22], mushroom compost [23], winterization oil cake [24], primary sludge from wastewater treatment plants [25], food waste [26], the organic fraction of municipal solid waste (OFMSW) [27], maize silage [28], dairy wastewater [29], palm oil mill wastewater [30], sugar industry wastewater [31], olive mill wastewater [24,32,33], glycerol [24,34], paper mill effluent [35], and hemp [36]. The focus of these works was the conversion of simple compounds or carbohydrates, and little attention was paid to more complex molecules like lipids. The most significant finding was related to the production of SCOAs using waste cooking oil as a substrate and an aerobic MMC as an inoculum. In this study, a maximum concentration of $7.14 \pm 0.201$ g COD/L of SCOAs was obtained, with propionic acid serving as the major acid produced, corresponding to 74% of the total SCOA amount, followed by valeric, butyric, and acetic acids [37].

Due to the popularity of fried food, waste cooking oils (WCOs) are produced worldwide, reaching volumes of 41 to 67 million tons every year [38–40]. The treatment of this waste is important to avoid contamination in water bodies and soil ecosystems, which can cause deoxygenation of water and a reduction in soil fertility [41]. Traditionally, WCOs are used for animal feed [42]. However, during the frying process, cooking oils can undergo hydrolysis, thermal oxidation, and polymerization, resulting in the production of toxic compounds, such as aldehydes (4-hydroxy-(E)-2-nonenal, (E)-4,5-epoxy-(E)-2-decenal, 4-hydroxy-(E)-2-hexenal, and 4-oxo-(E)-2-nonenal), peroxides, 1,4-dioxane, benzene, toluene, and hexylbenzene, making WCOs unsuitable for use in animal feed [43,44]. In recent years, the main valorization process for WCOs has been biodiesel production [42,45]. However, some compounds resulting from the frying of foods can affect biodiesel production, and the processing costs related to its purification are still high [42]. Due to their high chemical value and the environmental issues precipitated by their unsafe deposition, the valorization of WCOs was the aim of the present work. In this study, it was hypothesized that SCOA production through acidogenic fermentation could be achieved using industrial spent frying oil (SFO). For this purpose, SFO was characterized, and acidogenic experiments were performed using different sources of inocula, including MMCs from the anaerobic and aerobic tanks of a municipal wastewater treatment plant (WWTP) and aerobic sludge from the WWTP of a biodiesel production plant, as well as different food-to-microorganism (F/M) ratios.

## 2. Materials and Methods

### 2.1. Substrate

SFO was recovered using a Soxhlet (chloroform/methanol 2:1 *v/v*) extractor from brownish potato chip industry frying residue provided by A Saloinha, Lda. (Mafra, Portugal) [46]. Oleic acid was purchased from Sigma-Aldrich, Darmstadt, Germany.

### 2.2. Acidogenic Fermentation Analysis

#### 2.2.1. Inocula

The MMCs used as inocula were collected from the aerobic (AES) and anaerobic (ANS) tanks of the municipal WWTP SIMRia (Aveiro Sul, Portugal) and from the aerobic tank (BPS) of the WWTP of the biodiesel production company PRIO (Gafanha da Nazaré, Portugal), which should contain bacteria more adapted to complex lipid carbon sources. In addition, when using aerobic populations, there is a higher probability of finding more acidogenic bacteria than methanogenic ones, thus preventing the loss of SCOAs due to their conversion into methane [47,48].

#### 2.2.2. Experimental Set-Up

Batch tests were conducted in encapsulated flasks with 100 mL working volumes, maintained at 28 °C with constant stirring at 300 rpm, and purged with $N_2$ prior to incubation to ensure anaerobic conditions. SFO and oleic acid were used as carbon sources in a proportion of 1 gCOD of SFO and oleic acid per 1 gCOD of MMCs with the following mineral medium: 160 mg/L of $NH_4Cl$, 160 mg/L of $KH_2PO_4$, 80 mg/L of $CaCl_2 \cdot 2H_2O$, 160 mg/L of $MgSO_4 \cdot 7H_2O$, 800 mg/L of $NaHCO_3$, 200 mg/L of $CoCl_2 \cdot 6H_2O$, 30 mg/L of $MnSO_4 \cdot 7H_2O$, 10 mg/L of $CuSO_4 \cdot 5H_2O$, 100 mg/L of $ZnSO_4 \cdot 7H_2O$, 300 mg/L of $H_3BO_3$, 30 mg/L $(NH_4)_6Mo_7O_2 \cdot 4H_2O$, and 20 mg/L of $NiCl_2 \cdot 6H_2O$. Every day, 2.0 mL of a sample was collected under anaerobic conditions and centrifuged at $11,337 \times g$ for 10 min (MiniSpin, Eppendorf, Düsseldorf, Germany). The pellet was discharged, and the supernatant was stored at $-16$ °C for further determination of SCOAs. All the analyses were performed in single mode.

### 2.3. Analytical Methods

#### 2.3.1. Determination of SCOAs

A total of 800 µL of each sample was filtered using Vecta Spin Tubes (Whatman, Maidstone, UK) with a membrane of 0.2 µm (Whatman, Maidstone, UK) at $4293 \times g$ (MiniSpin Eppendorf, Düsseldorf, Germany) for 15 min before HPLC injection. SCOA concentrations in the acidogenic tests were measured in a Rezex ROA—Organic Acid $H^+$ (8%) column (Phenomenex, Torrance, CA, USA) at 65 °C and using a refractive index detector (Merck, Darmstadt, Germany), employing $H_2SO_4$ 0.005 N as an eluent (0.5 mL/min). The calibration curves were determined frequently using freshly prepared standards in the range of 0–3 g/L for SCOAs to ensure the method linearity.

#### 2.3.2. Chemical Oxygen Demand (COD)

COD was measured using Spectroquant Kit (Merck Millipore, Darmstadt, Germany), and the solutions used were prepared according to standard methods [49]: a digestive aqueous solution containing $K_2Cr_2O_7$, $HgSO_4$, and $H_2SO_4$ and an acid solution containing $H_2SO_4$ and $AgSO_4$ were prepared. To 2 mL of a properly diluted sample were added 1.2 mL of digestive solution and 2.8 mL of acid solution. The mixture was incubated at 150 °C for 2 h. After cooling, the absorbance at 600 nm was measured. Calibration was performed frequently using freshly prepared standards of glucose with COD concentrations between 0 and 1 g/L to ensure the method linearity.

#### 2.3.3. Fatty Acid Methyl Esters Determination

Fatty acid methyl esters (FAMEs) of SFO were determined, in triplicate, after alkaline-catalyzed transesterification and GC-FID analysis in a DB-FFAP column (30 m × 0.32 mm

(I.D.) × 0.25 μm film thickness, J&W Scientific Inc., Folsom, CA, USA) [46,50]. For this process, 1.5 mg of SFO and 1 mL of 1.5 mg/mL heptadecanoate methyl ester in *n*-hexane and 0.2 mL of KOH methanolic solution were used as the internal standard and the catalyst, respectively. The identification of the FAME compounds was based on the comparison of their retention times with those obtained using a commercial FAME mixture ($C_8$–$C_{24}$) standard.

### 2.3.4. pH Measurement

The pH of the samples was measured using an electrode, InPro 3030/200 (Mettler Toledo, Columbus, OH, USA), and a benchtop meter, sensION+ MM340 (Hach, Loveland, CO, USA), at 25 °C. Prior to the measurements, the pH meter was calibrated using technical buffer solutions of pH 4.00 ± 0.02, 7.00 ± 0.02, and 10.0 ± 0.02.

### 2.3.5. Fourier Transform Infrared (FTIR)

The FTIR analysis of substrates, inocula, and initial and final acidogenic samples was performed using a single-reflection diamond ATR system in Bruker spectrometer (Alpha Platinum-ATR, Bruker, Billerica, Massachusetts, USA). The FTIR spectra were captured at a resolution of 16 cm$^{-1}$ and with uni-directional scans, 32 co-added scans, and a wave number range of 4000 to 600 cm$^{-1}$ (mid infrared region) [46]. The FTIR analyses were performed, in triplicate, after the samples were dried via cold air circulation. To allow for a comparison between the different samples under study, each spectrum was normalized.

### *2.4. Calculations*

Concentrations were converted from g/L to g COD/L using conversion factors that represent the mass (g) of oxygen required to oxidize 1 g of compound based on the oxidation reactions for each compound [47]. The overall oxidation equation is presented in Equation (1):

$$\text{a compound} + \text{b } O_2 \rightarrow \text{c } CO_2 + \text{d } H_2O, \tag{1}$$

where a, b, c, and d represent the stoichiometric coefficients of the equation. The conversion factor (cf) was then calculated using Equation (2):

$$cf\left(\frac{gO_2}{g}\right) = \frac{b \times M(O_2)}{a \times M(\text{compound})}, \tag{2}$$

The conversion factors were 1.07 g $O_2$/g for glucose and acetic acid; 1.51 g $O_2$/g for propionic acid; 1.82 g $O_2$/g for butyric and isobutyric acids; and 2.04 g $O_2$/g for valeric acid [47].

The acidification degrees (ADs) were calculated for the fermentative process. The acidification degree represents the amount of substrate consumed to produce SCOAs. Substrate concentration was calculated theoretically for oleic acid and SFO. AD was determined using the maximum concentration of SCOAs for each experiment via employing Equation (3):

$$AD(\%) = \frac{[\text{SCOAs}]\text{MAX}}{\text{SubstrateTotalCOD}} \times 100, \tag{3}$$

Odd/even number SCOA ratio represents the ratio between the concentration of odd-number carbon SCOAs and even-number carbon SCOAs of a sample, where SCOA concentration was at its maximum for all experiments. Odd/even number SCOA ratio was determined using Equation (4):

$$\frac{\text{Odd}}{\text{Even}}\text{ratio} = \frac{[\text{OddnumbercarbonSCOAs}]}{[\text{EvennumbercarbonSCOAs}]}, \tag{4}$$

Peak assignment was performed automatically using injection method software. Integration and baseline corrections of obtained chromatograms were performed using HPLC

software. Peak height was exported to Excel, which was used to calculate SCOA concentrations by employing calibration curves of the injected standards.

## 3. Results and Discussion

### 3.1. Spent Frying Oil Characterization

Before the acidogenic fermentation experiments, the SFO was characterized to understand which carbon compounds were present and how much organic carbon was available. According to the FAME profile, it was found that SFO was mainly composed of oleic (81%), palmitic (11%), linoleic (6%), and stearic (2%) acids. This composition is normally found in vegetable cooking oils, namely, in olive oil [51]. Since the predominant fatty acid present in the analyzed SFO was oleic acid, a commercial standard of this acid was chosen for use as a model molecule for the acidogenic fermentation experiments.

The FTIR spectrum of SFO was acquired, and it was very similar to the spectrum of oleic acid (Figure 1). Both spectra presented adsorption bands at 2916 cm$^{-1}$ and 2851 cm$^{-1}$, which correspond to the long fatty acid chains' aliphatic regions, and at 1452 cm$^{-1}$, which corresponds to the terminal methyl (-CH$_3$) of the fatty acids. The oleic acid spectrum presented a notable adsorption band at 1703 cm$^{-1}$, while a band at 1740 cm$^{-1}$ was observed for SFO. These two bands correspond to the carboxylic terminal of oleic acid and the carbonyl group of the ester bond of SFO triacylglycerides, respectively [46,52].

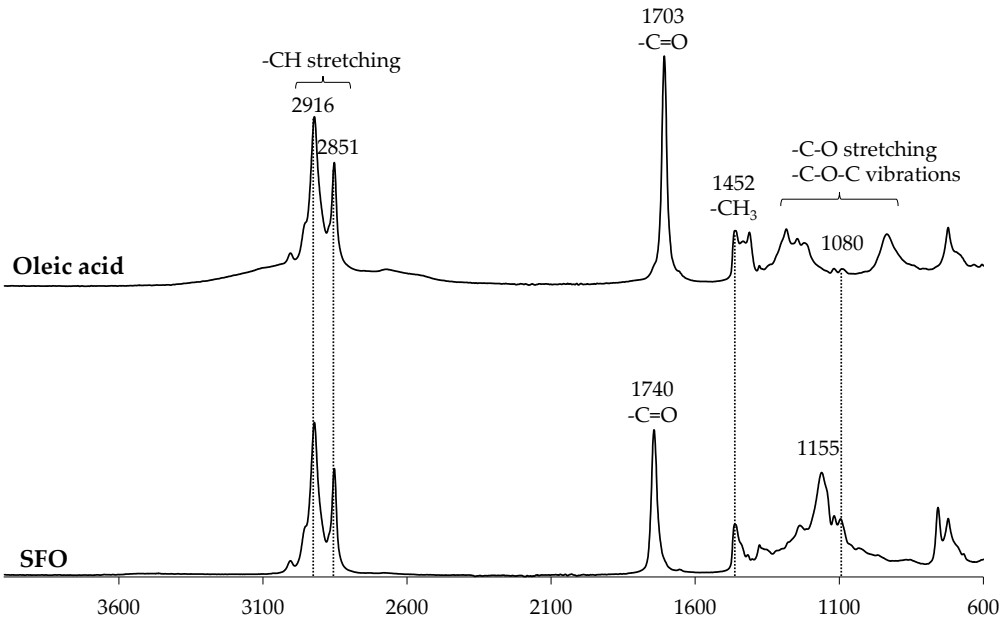

**Figure 1.** FTIR spectra of oleic acid and spent frying oil (SFO).

The amount of organic carbon in the SFO was quantified as COD, and a value of 1500.3 ± 839.3 g COD/L was obtained. The high standard deviation obtained can be related to the difficulty of applying the COD method to these high-fat and hydrophobic samples since this method involves the use of aqueous solutions in which the SFO and oleic acid substrates were not soluble. Efforts to minimize this issue were made by directly weighing the substrates instead of pre-solubilizing them.

### 3.2. Acidogenic Fermentation of Oleic Acid

Due to the complexity and recalcitrance of SFO, preliminary tests using oleic acid as a model molecule were performed. The aim of these experiments was to verify the capacity of the three selected MMCs to degrade a molecule with a hydrophobic portion. For this purpose, the MMCs from municipal WWTP, AES, and ANS were chosen as inocula due to their ability to degrade a wide range of organic compounds, particularly lipids like those present in SFO. ANS was tested because acidogenic fermentation is an anaerobic process.

On the other hand, the use of an aerobic inoculum in an anaerobic process could be a way to avoid the presence of methanogenic organisms that are strict anaerobes (in contrast, acidogens are facultative aerobes) [47,48]. In this way, the conversion of the produced SCOAs to methane could be prevented. A third inoculum was collected at the anaerobic tank of the WWTP of a biodiesel production plant that uses purified SFO as a raw material. In this microbial community, we expected to find microorganisms able to convert fatty acids like oleic acid into SCOAs.

As a starting point for these experiments, a food-to-microorganism ratio of 1 gCOD/gCOD (1:1) was chosen, and the determined COD values of the three inocula are shown in Table 1. The AES population showed the highest amount of organic matter, 7.2 ± 0.0 gCOD/L, and the ANS population showed a lower one (3.6 ± 0.0), while the BPS population, due to sample homogenization issues, presented a high standard deviation (4.1 ± 1.7 gCOD/L).

**Table 1.** COD values of the three inocula used in this work.

| Inoculum | COD (g/L) |
| --- | --- |
| AES | 7.2 ± 0.0 |
| ANS | 3.6 ± 0.0 |
| BPS | 4.1 ± 1.7 |

The three experiments lasted 28 days, and the evolution of the total concentration of SCOAs and pH, as well as the SCOA composition corresponding to the maximum concentration, are shown in Figure 2.

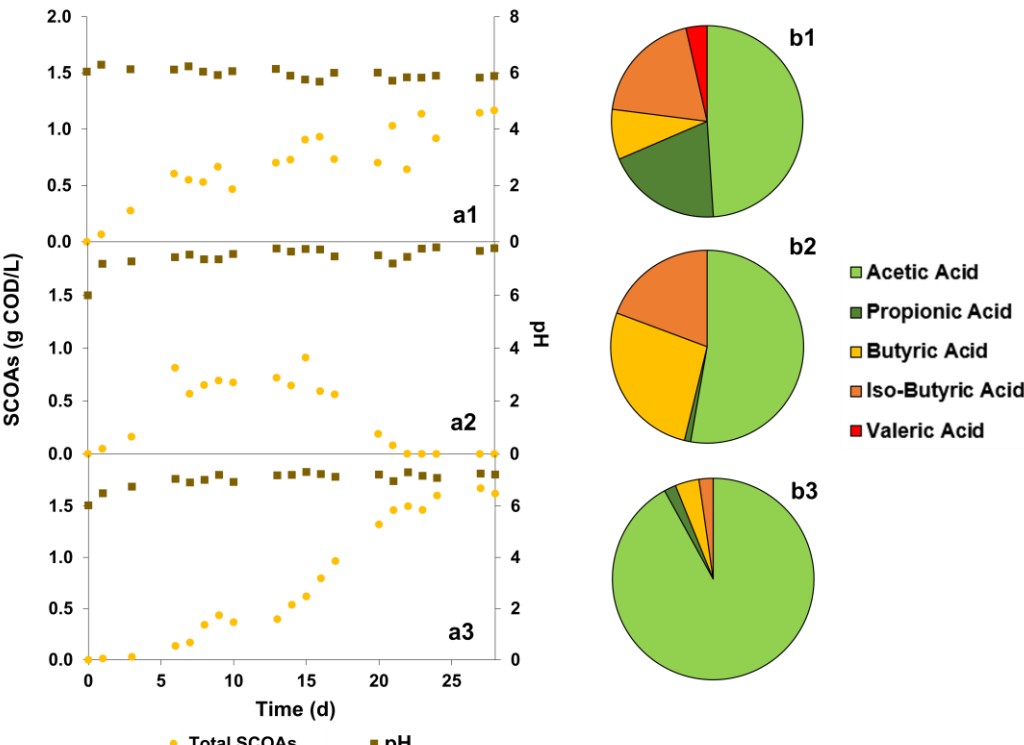

**Figure 2.** Evolution of SCOA concentration and pH (**a1**–**a3**) throughout the acidogenic fermentation of oleic acid experiments, and SCOA compositions (**b1**–**b3**) on the day a maximum value, expressed as percentage of molar basis, was achieved for each inoculum tested: AES: (**a1**,**b1**); ANS: (**a2**,**b2**); BPS: (**a3**,**b3**).

In all the experiments, the MMCs were able to produce SCOAs from oleic acid, with the experiment with the BPS inoculum resulting in the highest concentration of SCOAs, 1.67 gCOD/L. During the fermentation process, pH was not controlled, and since SCOAs were produced, the decrease in pH values observed throughout the operational period

was expected, but the changes in pH values in the three experiments were different. The AES experiment showed a stable pH of ca. 6.00 during the operational period, while the ANS and BPS experiments revealed an increase in pH on the first days, with the pH values increasing throughout the experiment, stabilizing between 7.50 and 8.00.

Due to the unavailability of sample amount required for FAME determination throughout the experiments, FTIR analysis of the initial and final samples was performed to verify oleic acid consumption. Figure S1 (Supplementary Material) shows the initial and final FTIR spectra for all the experiments. None of the final sample spectra exhibited a band at around 1703 cm$^{-1}$, which corresponds to the carbonyl group of oleic acid, leading to the conclusion that oleic acid was consumed.

For the AES and BPS experiments, SCOA production was gradual until days 28 and 27, respectively, when both experiments showed the maximum concentrations of SCOAs, while the ANS experiment led to the maximum SCOA concentration at day 15. Normally, SCOAs are biosynthesized from pyruvate (for the case of propionic acid) or acetyl-CoA (for the case of acetic and butyric acids) [13]. In the case of lipids, they undergo β-oxidation and are thus converted into SCOAs. It is possible to hypothesize that the complexity of oleic acid and the number of enzymatic steps and energy amount required for hydrolysis [53] led to the large amount of time required for conversion to SCOAs. From day 15 until day 24 for the ANS experiment, SCOAs were completely catabolized, possibly due to methanogenesis, which is corroborated by the fact that the pH range for methanogens growth is usually 6.5 to 8.2 [54], and the pH of the assay was around 7.50 from day 1 until the final day of operation. Since methanogens are strict anaerobes, while acidogenic bacteria are facultative anaerobes [47,48], this catabolism was not observed in the AES experiments. Since methane production was not quantified, it was not possible to conclude that methanogenesis was the result of the catabolism of SCOAs by the bacteria present in ANS. Further studies should evaluate the production of methane and the necessity of strategies for hindering methane production and the catabolism of SCOAs [55].

The AES experiment showed a higher diversity of SCOAs, with the production of acetic, propionic, butyric, iso-butyric, and valeric acids, with acetic acid (49%) being the dominant organic acid, followed by propionic acid (20%). The ANS experiment showed some diversity, with propionic acid produced in small amounts while valeric acid was not detected. Again, acetic acid (53%) was the dominant acid, followed by butyric acid (27%). On the other hand, the BPS experiment resulted in the production of notable amounts of acetic acid (92%), with vestigial production of other SCOAs (Figure 2). Rughoonundun et al. [56] used pretreated bagasse and mixed sewage sludge to perform acidogenic fermentation; at pH 7, the major acid produced was acetic acid. The AES experiment had a different SCOA distribution compared to that in the study performed by Wei et al. [57], which also used oleic acid as substrate for SCOA production. This study reported that butyric acid was the major acid produced, while the results obtained in the AES experiment showed an 8% proportion of butyric acid.

In terms of maximum SCOA concentration, the BPS experiment yielded the highest total concentration, 1.67 g COD/L, while the AES and ANS experiments resulted in lower maximum concentrations equal to 1.17 and 0.91 g COD/L. Despite this, the BPS experiment yielded the highest acidification degree, 55.67%, followed by such degrees reported in the ANS (36.40%) and AES (23.40%) experiments, leading to the conclusion that BPS can successfully hydrolyze fatty acid chains for SCOA conversion.

The Odd/Even ratio is a proportional measure of odd carbon SCOAs and even carbon SCOAs that is known to affect PHA composition when using SCOA mixtures as feedstock for producing bacteria [58]. While even carbon SCOAs favor the synthesis of hydroxybutyrate (HB), odd carbon SCOAs facilitate the synthesis of hydroxyvalerate (HV) [59]. The mixture of both types of SCOAs results in the production of copolymers of HB and HV, which are much more interesting from the process point of view than homopolymers of HB. One study reported the production of a copolymer consisting of 90% of HB and 10% of HV at up to 44.7 wt% of cell dry weight when feeding an MMC with a mixture of SCOAs

with an Odd/Even ratio of 52:48 [60]. Therefore, it is of utmost importance to obtain a balanced SCOA composition to maximize PHAs yields and HV production. In the AES experiment, an Odd/Even ratio of 0.30 was obtained, while the ANS and BPS experiments yielded ratios of 0.01 and 0.02, respectively (Table 2). With these results, it is possible to suggest that when applying the SCOAs obtained in the AES experiment as feedstock for PHA production, HB with low HV amounts will be obtained, while the SCOAs of the ANS and BPS experiments will result in the production of HB.

**Table 2.** Main results for each inoculum tested with oleic acid as a carbon source.

| Inoculum | Time$_{MAX}$ (d) | [SCOAs]$_{MAX}$ (g COD/L) | AD (%) (%COD/COD) | Odd/Even Ratio (mol/mol) |
|---|---|---|---|---|
| AES | 28 | 1.17 | 23.40 | 0.30 |
| ANS | 15 | 0.91 | 36.40 | 0.01 |
| BPS | 27 | 1.67 | 55.67 | 0.02 |

Since all the populations were able to convert oleic acid into SCOAs, despite the differences in the composition, the three of them were tested as inocula in the acidogenic experiments with SFO.

### 3.3. Acidogenic Assays of Spent Frying Oil

3.3.1. 1:1 Food-to-Microorganism (F/M) Ratio

As a starting point for the assays with SFO, the same F/M tested with oleic acid was chosen, namely, 1 gCOD/gCOD (1:1). The three experiments lasted 28 days, and the evolution of the total concentration of SCOAs and pH, as well as the SCOA composition corresponding to the maximum concentration, are shown in Figure 3. The main results of all the experiments are described in Table 3.

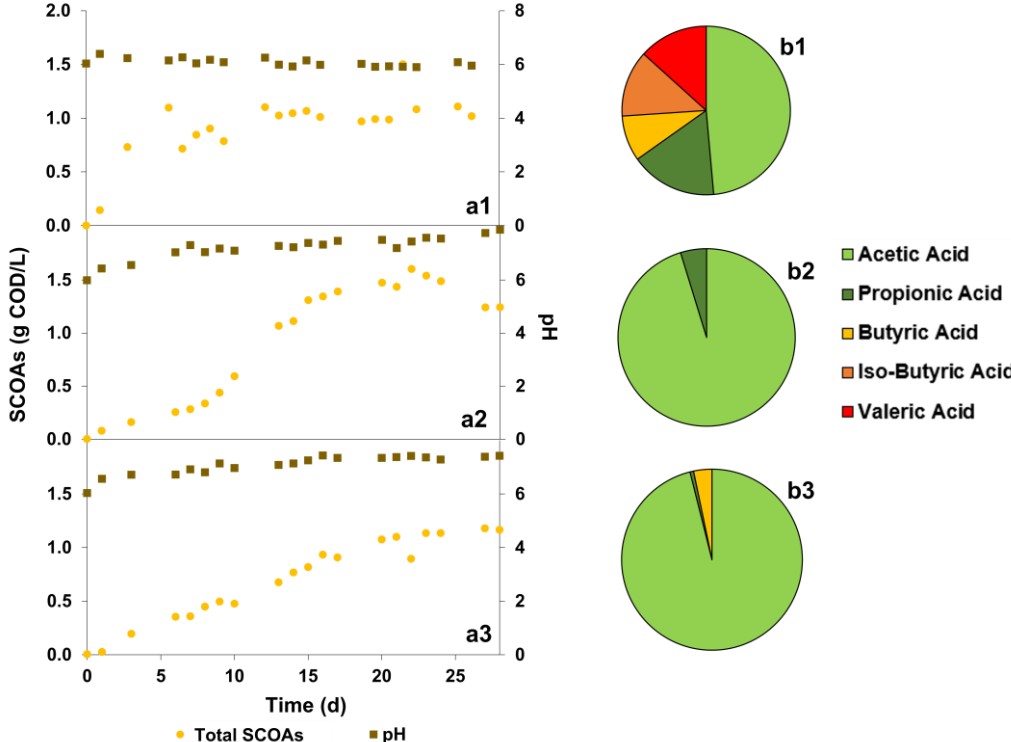

**Figure 3.** Evolution of SCOA concentration and pH (**a1**–**a3**) throughout the acidogenic-fermentation-of-SFO experiments and SCOA composition (**b1**–**b3**) on the day the maximum value, expressed as percentage of molar basis, was achieved for each inoculum tested: AES: (**a1**,**b1**); ANS: (**a2**,**b2**); BPS: (**a3**,**b3**).

**Table 3.** Main results for each inoculum tested with SFO as carbon source.

| F/M Ratio (g COD/g COD) | Inoculum | $\text{Time}_{\text{MAX}}$ (d) | $[\text{SCOAs}]_{\text{MAX}}$ (g COD/L) | AD (%) (%COD/COD) | Odd/Even Ratio (mol/mol) |
|---|---|---|---|---|---|
| 1:1 | AES | 23 | 1.50 | 30.00 | 0.43 |
| | ANS | 22 | 1.60 | 64.00 | 0.00 |
| | BPS | 27 | 1.18 | 39.33 | 0.01 |
| 2:1 | AES | 15 | 0.80 | 8.00 | 0.58 |
| | BPS | 14 | 0.32 | 5.33 | 0.23 |
| 1:2 | AES | 23 | 0.60 | 24.00 | 0.15 |
| | BPS | 17 | 0.18 | 12.00 | 0.31 |

During the SFO fermentation operational period, the AES experiment displayed stable pH values of ca. 6.0, while the ANS and BPS experiments exhibited an increase in pH throughout the experiment from ca. 6.0 to values around 7.5, with the ANS experiment reaching the maximum value of ca. 7.9 on the last day of the fermentation process (Figure 3). This behavior followed the trend observed for the experiments performed with the simple molecule of oleic acid (Figure 2). This buffering activity was already observed during the acidification of complex substrates like sulfite spent liquor, a complex waste from the pulp and paper industry, and spent coffee grounds from coffee processing, respectively [20,47].

As was conducted for oleic acid, FTIR analysis was performed on the initial and final fermentation samples, and it was possible to see differences between the spectra that confirmed that the consumption of SFO fatty acids occurred (Figure S2, Supplementary Material). The initial FTIR spectra of the AES and ANS exhibited a slight adsorption band at around 1740 cm$^{-1}$, which is characteristic of carbonyl groups with an ester bond, corresponding to the carbonyl groups of SFO triacylglycerides [46], which are not present in the final samples' spectra. Furthermore, the AES and ANS initial spectra exhibit slight adsorption bands at around 2916 cm$^{-1}$ and 2851 cm$^{-1}$, which are characteristic of -CH stretching, corresponding to the long aliphatic chains of the SFO triacylglycerides, which were absent in the final samples' spectra, leading to the conclusion that SFO was totally consumed.

For the AES experiment, SCOAs were produced in significant quantities until day 6, deaccelerating until day 23, when the maximum SCOA concentration of the experiment, 1.50 g COD/L, was achieved, remaining stable until the end of the experiment. The ANS experiment exhibited gradual SCOA production throughout the duration of the experiment until a maximum concentration was achieved on day 22, namely, 1.60 g COD/L. After day 22, SCOAs were slightly consumed as observed in the experiment with oleic acid (Figure 2). As explained earlier, this behavior is probably due to methanogenesis. Since methanogenesis pH range is from 6.5 to 8.2 [55], and as the pH values of the ANS experiment varied in this range, it can be assumed that ANS had active methanogens that redirected the produced SCOAs to methane production. On the other hand, in the experiment with BPS, the production of SCOAs occurred until the end of the experiment, reaching the maximum concentration on day 27 with 1.18 g COD/L, the lowest value of the three experiments. For the experiments with AES and BPS, the maximum concentration of SCOAs was achieved faster when compared to the experiment with oleic acid (Table 3), while the microbial population of ANS needed a longer time to achieve the maximum SCOA concentration. This longer time to produce SCOAs was probably due to the complexity of SFO when compared to oleic acid, requiring more enzymatic steps and energy to fully hydrolyze SFO to SCOAs.

The SCOA composition obtained in the AES experiment was like the one observed when using oleic acid in the same F/M (Figure 2), with the production of acetic, propionic, butyric, iso-butyric, and valeric acids at a 48:17:9:13:13% molar basis, respectively. The experiments with ANS and BPS allowed the production of lower SCOA diversity than that in the experiments with oleic acid (Figure 3). Ping et al. [37] conducted an acidogenic

fermentation study using waste cooking oil and aerobic sludge. In their study, all the experiments allowed the authors to obtain a major concentration of propionic acid, with minor concentrations of valeric, acetic, and butyric acids. Although the AES experiment allowed the production of a balanced concentration of propionic acid, it was not the major acid of the experiment. This difference in results can be attributed to the differences in the microbial populations in the waste streams of the inocula used, despite both being aerobic WWTP sludges. Further studies should include the characterization of the microbial communities of the populations used as an inoculum to relate it with the diversity of the SCOAs obtained [61].

In terms of acidification degree, the ANS experiment led to the highest value, amounting to 64.00%, followed by the BPS (39.33%) and AES (30.00%) experiments. Despite the high acidification degree reported in the ANS experiment, due to the catabolism effect observed and the low diversity of SCOAs obtained, it was decided that ANS would not be used in further experiments.

Regarding the Odd/Even ratios (Table 3), these were quite like the 1:1 F/M ratio of the oleic acid experiments (Table 2). The AES experiment allowed for the acquisition of a ratio of 0.43, while the ANS and BPS experiments allowed for the acquisition of ratios of 0.00 and 0.01, respectively. With these results, it can be suggested that the fermented streams obtained in the AES experiment, when applied to the production of PHAs as feedstocks, will probably result in a moderate production of HV and HB, while the fermented streams obtained in the ANS and BPS experiments will probably result in the production of HB.

To assess the effect of changing the F/M ratio by changing the amount of SFO used, two experiments, 2:1 F/M and 1:2 F/M, were performed.

### 3.3.2. F/M Ratios of 2:1 and 1:2 with SFO

The two F/M ratios of 2:1 and 1:2 with SFO were tested with the AES and BPS inocula, and the main results of these experiments are shown in Table 3.

To assess SFO consumption in all the experiments, once again, FTIR was performed on the initial and final samples of the experiment (Figures S3 and S4, Supplementary Materials). All the initial spectra exhibited a slight adsorption band at around 1703 cm$^{-1}$, which is characteristic of carbonyl groups, probably corresponding to free fatty acids of SFO. Despite this band being lower in the final spectra of the samples, it was still present, leading to the conclusion that SFO was only partially consumed.

The maximum SCOA concentrations of all the experiments were lower when compared to the results obtained in the 1:1 F/M ratio experiments (Table 3). These lower concentrations were probably the result of substrate inhibition in the case of 2:1 F/M and substrate limitation in 1:2 F/M for both inocula. In the case of 2:1 F/M, long-chain fatty acids could have had an inhibitory effect on acidogenic fermentation due to adhesion to the cell walls, which could lead to a decrease in nutrient transportation and mass transfer limitations [62–68]. The undissociated fatty acids of SFO probably induced this inhibition, resulting in lower concentrations. Moreover, the acidification degrees were also lower when compared to the ones obtained in the 1:1 F/M ratio experiments with SFO (Table 3). The AES experiment resulted in an acidification degree of 8.00%, while the BPS experiment exhibited a lower value (5.33%). Despite the lower SCOAs yields, these results lead to the conclusion that AES is more capable of triacylglyceride hydrolysis than BPS, having higher SCOAs yields and distributions. This is probably due to AES being from a municipal WWTP, in which there is a larger variety of organic compounds, leading to a better adaptation of AES to a variety of different wastes and a broader metabolic potential than BPS, which is a byproduct of biodiesel production effluent, being more selected and adapted to the conversion of glycerol into SCOAs.

The Odd/Even ratios (Table 3) obtained were increased when compared to the 1:1 F/M ratio experiments. The AES experiment led to an Odd/Even ratio of 0.58, while the BPS experiment allowed for the acquisition of an Odd/Even ratio of 0.23. It seems that higher concentrations of lipids lead to a higher diversity of SCOAs produced. These results

allow us to suggest that applying these fermented streams to the production of PHAs as feedstocks could result in the production of HB and HV, with AES leading to the production of higher HV amounts.

The AES experiment resulted in a better SCOA production yield, but it was still lower than that observed with 1:1 F/M, with an acidification degree of 24.00%, while the BPS experiment resulted in an acidification degree of 12.00% (Table 3). This can lead to the conclusion that a 1:1 F/M ratio allowed for the best SCOA results and yields among the tested ratios. Not only that, but AES resulted in a more diverse mixture of acids despite having the worst Odd/Even ratio result with 1:2 F/M. The AES experiment yielded a low Odd/Even ratio of 0.15, while the BPS experiment yielded a higher Odd/Even ratio of 0.31. These results suggest that, when applying these fermented streams as feedstocks for PHA production, the fermented streams obtained in the experiment with AES will probably result in major production of HB, with very low amounts of HV, while the ones obtained in the experiment with BPS will potentiate the production of HV. These results show that SFO can be used as a substrate for acidogenic fermentation with these MMCs to produce SCOAs with a somewhat balanced structural distribution, constituting an interesting topic of study for future acidogenic fermentation studies using lipid-rich substrates in accordance with circular economy and sustainability concepts.

## 4. Conclusions

From the results of this work, it is possible to conclude that SFO can be applied in acidogenic fermentation, leading to the production of a high diversity of SCOAs, with the 1:1 F/M ratio experiments leading to the highest acidification degrees obtained in all the F/M ratios studied. Moreover, when using SFO, the inocula seem to have a great impact on SCOA production and diversity, with the AES populations being more efficient for SCOA production using lipid-based substrates, such as SFO, than the ones from crude glycerol production (BPS). Due to the scarcity of studies using lipidic substrates, this study can be considered a foundation for future acidogenic fermentation studies following circular economy concepts. As future work, acidogenic fermentation should be tested on a larger scale, for example, on a moving-bed batch reactor (MBBR), for the future study of applications in PHA production with mixed microbial cultures.

**Supplementary Materials:** The following supporting information can be downloaded at https://www.mdpi.com/article/10.3390/fermentation9110975/s1. Figure S1. FTIR spectra of initial and final samples of the experiment with a 1:1 F/M ratio of acidogenic fermentation of oleic acid; Figure S2. FTIR spectra of initial and final samples of the experiment with a 1:1 F/M ratio of acidogenic fermentation of SFO; Figure S3. FTIR spectra of initial and final samples of the experiment with a 2:1 F/M ratio of acidogenic fermentation of SFO; Figure S4. FTIR spectra of initial and final samples of the experiment with a 1:2 F/M ratio of acidogenic fermentation of SFO.

**Author Contributions:** Conceptualization, A.O., S.P. and L.S.S.; methodology, A.O.; writing—original draft preparation, A.O.; writing—review and editing, A.O., S.P. and L.S.S. All authors have read and agreed to the published version of the manuscript.

**Funding:** Thanks are due to the FCT/MCTES for their financial support provided to the CICECO-Aveiro Institute of Materials (FCT Ref. UIDB/50011/2020, UIDP/50011/2020, and LA/P/0006/2020) and LAQV-REQUIMTE (UIDB/50006/2020, UIDP/5006/2020) research units and CQ-VR at UTAD Vila Real (UIDP/00616/2020), through PT national funds, and when applicable, co-financed by FEDER, within the PT2020 Partnership Agreement and Compete 2020. The FCT/MCTES is also thanked for the post-doc grant SFRH/BPD/117213/2016 (SP). CICECO is also thanked for research grant EdtIB.41-CICECO/2023 UIDB/50011/2020 &UIDP/50011/2020 (AO). This work is also supported by NORTE 2020, under the PT 2020 Partnership Agreement, through the ERDF and FSE.

**Institutional Review Board Statement:** Not applicable.

**Informed Consent Statement:** Not applicable.

**Data Availability Statement:** Data are contained within the article and Supplementary materials.

**Acknowledgments:** The authors also thank to "A Saloinha, Lda." for providing the potato byproducts used to recover the spent frying oil.

**Conflicts of Interest:** The authors declare no conflict of interest.

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
