# Peer review of "Employing Spent Frying Oil as a Feedstock to Produce Short-Chain Organic Acids Using Mixed Microbial Cultures"

_fermentation, doi:10.3390/fermentation9110975_

Round 1

Reviewer 1 Report

Comments and Suggestions for Authors

Major points:

1)      One confusing aspect of this contribution is the mixed use of “COD” and concentrations for acids.  One problem is that the “COD” value of each organic acid is not equivalent (larger acids have higher CODs).  Thus, the COD is weighted by larger acids.  If acetic acid happens to be converted to propionic acid, for example, then the COD might increase, while the opposite would occur if propionic acid were converted to acetic acid.  Also, the use of both results in awkward presentations.  For example, Figure 2 shows CODs on the left panel, while it shows percentage of each acid on the right panel, presumably based on mass fraction.  My recommendation is that ALL data be presented in mass units.  In other words, the concentrations of acids be reported in “mg/L” throughout.

2)      The authors allude to “PHA” production in the work (e.g., lines 16, 23, 25, 43, 273, 281, 284, 358, 361, 394, 402, 418), as well as “HB”.  However, at no time was PHA generated.  Therefore, any statements regarding PHA are out of scope and should be removed.

3)      There is a significant difference between the three inocula examined.  In particular, the AES generated the least acetic acid, and the most propionic acid, both from oleic acid and from SFO.  One important conclusion is that the inoculum makes a significant difference in the distribution of acids as an outcome.  This conclusion is critical to the scientific community because it addresses reproducibility.  Stated another way, other researchers might not be able to reproduce any of the results described herein because the inocula are so different.  If this is the case, then what is the value of this contribution which cannot readily be reproduced?  The contribution would be significantly improved by identifying the microbes in the various inocula, which would at least allow future researchers the possibility of reproducing these results.

Other minor comments:

Iine 18-19:  “After the characterization of SFO that showed that is mainly…” is not a complete sentence and also is missing an “it”.

Line 20:  I recommend using “MMCs” as the abbreviation for mixed microbial cultures, which refers to multiple cultures, so as to be consistent with the definition of PHAs and SCOAs.

The abstract should state some of the more important results and not merely be a description of what was done or the motivation for the work.

Line 24:  Not clear what “standing point” means.  Perhaps the authors mean “starting point”

Line 89:  Text contains a starting parenthesis “(“ without an ending one.  Perhaps a colon could be placed here.

Line 104-113:  The molybdate “6” should be a subscript.  Most of the salts listed have stable waters of hydration, and use of anhydrous salts as indicated would be exceedingly unusual.  The authors should make sure that the mass concentrations used reflect the correct concentrations (i.e., of the anhydrous or hydrated salts).

Line 111:  centrifugation should be reported in g-force not rpm (e.g., 5000 x g).

Line 185:  To improve the readability, I suggest not abbreviating oleic acid (OL).

Line 196:  The term “COD” has been defined, and then “COOD” is used.  Presumably the “COOD” should be changed to “COD”.  Also, the error on the measurements is 25-50%, which renders them useless.  The authors either must find a useful method of making these measurements, use some other measurable parameter, or not report these values.  Having such a significant relative error calls into question the correctness of other reported COD measurements.  Clearly the samples have to be diluted sufficiently for the measurement.  Also, oleic acid is a pure compound and its “COD” should be readily calculable.  Specifically, the complete oxidation of oleic acid is:

C18H34O2 + 25.5O2 à 17H2O + 18CO2.

The oxidation of oleic acid therefore requires 25.5 moles O2/mole oleic acid.  The molar mass of oleic acid is 282.468 g/mol and the density is 0.895 g/mL, while the molecular weight of oxygen is 32 g/mol.  Thus:

(25.5 moles O2/mole oleic acid) x (32 g O2/mole O2) x (mole oleic acid/282.468 g oleic acid) x (0.895 g oleic acid/mL) x (1000 mL/L) = 2,585 g/L

Line 201:  “pre” should not have an accent on the “e”

Line 208:  “an anaerobic process” instead of “a anaerobic process”

Line 209:  sentence starting “Preventing the presence of…” does not make sense.  Specifically, the word “signify” is probably not the intended word.

Line 215:  delete “were determined and”

Table 1:  The COD is reported to the nearest tenth but the error to the nearest one-hundredth.  The error should be the same significance as the measurement.  In other words, please report “7.2 +/- 0.0”, “3.6 +/- 0.0” and “4.1 +/- 1.7”.  Also in the text (line 216)

Line 216:  A comma should appear after “gCOD/L”

Figure 2: The values on the x-axis should be more consistent.  Specifically, the distance between 0 and 5 is much greater than the gaps between the other values, although each should be equidistant.  Not that the number of ticks between, for example, there are three ticks between “20” and “25”, suggesting each tick, awkwardly, represents 1.25 days.

Figure 2 panel b:  It is unclear whether the pie charts show mole fractions or mass fractions.

Line 238:  Remove the word “fully”.  To be fully consumed, the oleic acid would have been converted entirely to carbon dioxide, which clearly was not accomplished.

The authors used the word “assay” indiscriminately, as in for example an adjective for AES, ANS or BPS.  The word has a very specific meaning, typically used for a specific analytical test for activity.  I recommend essentially eliminating this word throughout the document.  Where appropriate the word “experiment” or “inoculum” seem more correct.

Line 328:  “the” instead of “de”

Line 414:  Not clear what “Wi” means.

Comments on the Quality of English Language

Inaccurate or imprecise words are occasionally used.

Author Response

The authors thank Reviewer #1 for the insightful comments that contributed to the correction and a significant improvement of this article. All changes were made directly in the document, highlighted by the "track change" tool of MS Word.

Reviewer #1

Reviewer comment:

“Comments and Suggestions for Authors

Major points:

1) One confusing aspect of this contribution is the mixed use of “COD” and concentrations for acids.  One problem is that the “COD” value of each organic acid is not equivalent (larger acids have higher CODs).  Thus, the COD is weighted by larger acids.  If acetic acid happens to be converted to propionic acid, for example, then the COD might increase, while the opposite would occur if propionic acid were converted to acetic acid.  Also, the use of both results in awkward presentations.  For example, Figure 2 shows CODs on the left panel, while it shows percentage of each acid on the right panel, presumably based on mass fraction.  My recommendation is that ALL data be presented in mass units.  In other words, the concentrations of acids be reported in “mg/L” throughout.”

Authors answer:

Chemical oxygen demand (COD) is a technique widely used by researchers working in the field of biological conversion of complex waste. In this work it was used to quantify the total amount of carbon present in SFO, as well as in the inoculums’ biomass. As mentioned in the materials and methods section, this methodology was defined in the Standard Methods for the Examination of Water and Wastewater from the American Water Works Association. The fatty acid methyl esters composition of SFO, by FAME, was not enough to determine the total amount of carbon present in SFO since other compounds will be present deriving from the frying process. So, it is not possible to convert the results in gCOD/L in g/L. As stated in the materials and methods, SCOAs were quantified by HPLC and then the concentrations in g/L were converted to gCOD/L in accordance with the oxidation equation of each acid. In this way, we have all the values with coherent units, and we can determine the degree of acidification.

Regarding Figure 2, the reviewer is correct, the lack of information regarding units made it difficult to understand, thus the missing information was added in the caption.

Reviewer comment:

“2) The authors allude to “PHA” production in the work (e.g., lines 16, 23, 25, 43, 273, 281, 284, 358, 361, 394, 402, 418), as well as “HB”.  However, at no time was PHA generated.  Therefore, any statements regarding PHA are out of scope and should be removed.”

Authors answer:

The references to PHA are not clear because a brief explanation was missing in the Introduction. The main application of the mixtures of SCOAs obtained in the acidogenic fermentation of wastes is usual PHAs production by mixed microbial cultures, since there is no need to separate and purify the acids obtained. The current work is the first step of the three-step process that has the main goal of converting SFO into PHAs. This is the reason why the variety of SCOAs and the odd-to-even ratios are so important for the discussion of this work, because the variety and the presence of SCOAs with an odd number of carbon atoms will allow producing PHAs with different monomers, which is more interesting from the application point of view.

In the introduction a couple of sentences briefly explaining the relationship between SCOAs and PHAs were introduced with two references that show this connection.

Reviewer comment:

“3) There is a significant difference between the three inocula examined.  In particular, the AES generated the least acetic acid, and the most propionic acid, both from oleic acid and from SFO.  One important conclusion is that the inoculum makes a significant difference in the distribution of acids as an outcome.  This conclusion is critical to the scientific community because it addresses reproducibility.  Stated another way, other researchers might not be able to reproduce any of the results described herein because the inocula are so different.  If this is the case, then what is the value of this contribution which cannot readily be reproduced?  The contribution would be significantly improved by identifying the microbes in the various inocula, which would at least allow future researchers the possibility of reproducing these results.”

Authors answer:

The reason for testing three different inoculums was because we were expecting exactly different behaviors regarding SCOAs production. The justification for this was included in the materials and methods section. The different behaviors do not mean that the process is not reproducible at all. It means that the most suitable population for the production of SCOAs from SFO should come from an aerobic process and used to a higher variety of substrates, like the inoculum AES. Moreover, these are preliminary results since the main objective was to verify if it was possible to acidify SFO to obtain SCOAs and then use them for PHAs production.

Regarding the microbial composition of the populations, this will be the object of study in the future by using different molecular techniques, as it is usual in MMC processes, and might help to stablish a relation between cultures composition and the diversity of the SCOAs obtained. This idea was now included in the revised version of the manuscript.

Reviewer comment:

Other minor comments:

Iine 18-19:  “After the characterization of SFO that showed that is mainly…” is not a complete sentence and also is missing an “it”.

Line 20:  I recommend using “MMCs” as the abbreviation for mixed microbial cultures, which refers to multiple cultures, so as to be consistent with the definition of PHAs and SCOAs.”

Authors answer:

The corrections were made according to the reviewer suggestion.

Reviewer comment:

The abstract should state some of the more important results and not merely be a description of what was done or the motivation for the work.

Authors answer:

According to the reviewer suggestion, the most relevant results of the work were stated in the abstract.

Reviewer comment:

“Line 24:  Not clear what “standing point” means.  Perhaps the authors mean “starting point”

Line 89:  Text contains a starting parenthesis “(“ without an ending one.  Perhaps a colon could be placed here.”

Authors answer:

The corrections were made according to the reviewer suggestion.

Reviewer comment:

Line 104-113:  The molybdate “6” should be a subscript.  Most of the salts listed have stable waters of hydration, and use of anhydrous salts as indicated would be exceedingly unusual.  The authors should make sure that the mass concentrations used reflect the correct concentrations (i.e., of the anhydrous or hydrated salts).

Authors answer:

The “6” in molybdate was formatted as subscript. Also, the list of salts was reviewed and some of them were corrected, since the waters of hydration were missing, although the concentrations were correctly determined.

Reviewer comment:

Line 111:  centrifugation should be reported in g-force not rpm (e.g., 5000 x g).

Line 185:  To improve the readability, I suggest not abbreviating oleic acid (OL).

These two corrections were made according to the reviewer’s suggestion.

Reviewer comment:

“Line 196:  The term “COD” has been defined, and then “COOD” is used.  Presumably the “COOD” should be changed to “COD”.  Also, the error on the measurements is 25-50%, which renders them useless.  The authors either must find a useful method of making these measurements, use some other measurable parameter, or not report these values.  Having such a significant relative error calls into question the correctness of other reported COD measurements.  Clearly the samples have to be diluted sufficiently for the measurement.  Also, oleic acid is a pure compound and its “COD” should be readily calculable.  Specifically, the complete oxidation of oleic acid is:

C18H34O2 + 25.5O2 à 17H2O + 18CO2.

The oxidation of oleic acid therefore requires 25.5 moles O2/mole oleic acid.  The molar mass of oleic acid is 282.468 g/mol and the density is 0.895 g/mL, while the molecular weight of oxygen is 32 g/mol.  Thus:

(25.5 moles O2/mole oleic acid) x (32 g O2/mole O2) x (mole oleic acid/282.468 g oleic acid) x (0.895 g oleic acid/mL) x (1000 mL/L) = 2,585 g/L”

Authors answer:

The COOD was a typo and was corrected to COD. We agree that the error is high, but the measurement is not useless because so far is the only one possible since the hydrophobic origin of the substrate interfere with the method- Currently, several modifications of the method are being studied. Moreover, the high error does not hinder the main findings of the manuscript: it is possible to acidify SFO and a variety of SCOAs can be obtained, opening the door to develop a three-step process for PHAs production.

Regarding the oleic acid the value of COD used was calculated considering the oxidation reaction and not the measurement reported in the text done to determine how deviated was the method for a hydrophobic molecule. For this reason, the COD measurement of oleic acid was deleted from the text.

Reviewer comment:

“Line 201:  “pre” should not have an accent on the “e”

Line 208:  “an anaerobic process” instead of “a anaerobic process”

Line 209:  sentence starting “Preventing the presence of…” does not make sense.  Specifically, the word “signify” is probably not the intended word.

Line 215:  delete “were determined and”

Table 1:  The COD is reported to the nearest tenth but the error to the nearest one-hundredth.  The error should be the same significance as the measurement.  In other words, please report “7.2 +/- 0.0”, “3.6 +/- 0.0” and “4.1 +/- 1.7”.  Also in the text (line 216)

Line 216:  A comma should appear after “gCOD/L”

Authors answer:

All the corrections were made according to the reviewer’s suggestion.

Reviewer comment:

Figure 2: The values on the x-axis should be more consistent.  Specifically, the distance between 0 and 5 is much greater than the gaps between the other values, although each should be equidistant.  Not that the number of ticks between, for example, there are three ticks between “20” and “25”, suggesting each tick, awkwardly, represents 1.25 days.

Figure 2 panel b:  It is unclear whether the pie charts show mole fractions or mass fractions.

Authors answer:

The reviewer was right, the x-axis of Figure 2 was not correct and a new Figure with the x-axis correct was introduced. Also, the information about the pie charts showing mole fractions was added to the caption of this Figure.

“Line 238:  Remove the word “fully”.  To be fully consumed, the oleic acid would have been converted entirely to carbon dioxide, which clearly was not accomplished.

The authors used the word “assay” indiscriminately, as in for example an adjective for AES, ANS or BPS.  The word has a very specific meaning, typically used for a specific analytical test for activity.  I recommend essentially eliminating this word throughout the document.  Where appropriate the word “experiment” or “inoculum” seem more correct.

Line 328:  “the” instead of “de”

Line 414:  Not clear what “Wi” means.

Comments on the Quality of English Language Inaccurate or imprecise words are occasionally used.”

Authors answer:

The corrections were made according to the reviewer’s suggestion. Moreover, the manuscript was reviewed in detail for English language and style.

Reviewer 2 Report

Comments and Suggestions for Authors

In this study, the authors proved that spent frying oil can be used for short chain organic acids production by mixed microbial cultures.  Different MMCs from WWTP, AES, and ANS were tested and different F/M ratios were compared. From the results, it was concluded that AES had the better SCOAs production yield with 1:1 F/M. Some concerns were listed as below:

1, During the process, the pH was stable while SCOAs produced, why?

2, I suggest to test the methane production to prove the inference.

3, Different microbe diversity in different MMCs may be affect the diversity of SCOAs. I suggest to verify it.

4, Is the SCOAs production can be directly used for PHAs fermentation? If not, what extra steps should be taken before this? Is it economical?

5, 3.4.2 to 3.3.2

Author Response

The authors thank Reviewer #2 for the insightful comments that contributed to the correction and a significant improvement of this article. All changes were made directly in the document, highlighted by the "track change" tool of MS Word.

Reviewer #2

Reviewer comment:

“In this study, the authors proved that spent frying oil can be used for short chain organic acids production by mixed microbial cultures.  Different MMCs from WWTP, AES, and ANS were tested and different F/M ratios were compared. From the results, it was concluded that AES had the better SCOAs production yield with 1:1 F/M. Some concerns were listed as below:

1, During the process, the pH was stable while SCOAs produced, why?”

Authors answer:

Spent frying oil (SFO) is a very complex substrate that has triglycerides as main constituents, but probably there are other components also present in the original oil or resulting from the frying process that could promote some buffering activity. This is not the first time that this buffering activity was observed during the acidification of complex substrates. Queirós et al. (Fermentation 2017, 3, 20; doi:10.3390/fermentation3020020) and Pereira et al. (Biomolecules 2022, 12, 1284. doi.org/10.3390/biom12091284) observed a similar behavior during the acidification of sulfite spent liquor a complex waste from the pulp and paper industry and spent coffee grounds, respectively. This comment and the references were added to the text. Also, the stability of pH values could also be a consequence of the metabolism of the microorganisms. Future experiments will allow us to clarify what was causing the stability of pH.

Reviewer comment:

“2, I suggest to test the methane production to prove the inference.”

Authors answer:

To test the methane production during the acidification process is something to be done in the future since these are preliminary results. The aim of this work was only to verify the potential of SFO to produce SCOAs through acidogenic fermentation. Nevertheless, the idea of testing aerobic inoculums in an anaerobic process was also an effort to inhibit methanogenesis since most of methanogens are strict anaerobes, while the acidogens are facultative anaerobes and more likely to be found in those inoculums. This explanation was included in the Materials and Methods section of the revised version of the manuscript. However, there is still a lot of work to be done regarding the process optimization. This includes the measurement of methane production during the process since it will consume part of the SCOAs produced. If it is produced in a considerable amount, strategies that prevent the production of methane need to be studied. The text regarding the possibility of SCOAs catabolism to produce methane was completed according to this comment.

Reviewer comment:

“3, Different microbe diversity in different MMCs may be affect the diversity of SCOAs. I suggest to verify it.”

Authors answer:

The results of this work confirmed this finding, since three different populations were tested and different degrees of variety of the SCOAs were obtained from SFO. Another aspect to be clarified in the future is the microbial composition of the populations used as inoculum. FISH, DGGE and NGS techniques will allow to characterize the microbial populations in the inoculum and their evolution during the process and relate it with the type of SCOAs produced. A small comment on the necessity of characterization of the microbial communities was added to the text.

Reviewer comment:

“4, Is the SCOAs production can be directly used for PHAs fermentation? If not, what extra steps should be taken before this? Is it economical?”

Authors answer:

The main application of SCOAs obtained in the acidogenic fermentation of waste is usually for PHAs production by mixed microbial cultures (MMCs). In the introduction a couple of sentences briefly explaining the relationship between SCOAs and PHAs were introduced with two references that show the connection between SCOAs and PHAs production. In addition, as reported in several papers published in the last 15 years, the fermented stream enriched in SCOAs obtained in the acidification of a waste is quite easy to be used as substrate for PHAs production by MMC. There is only the need to remove the biomass by centrifugation or filtration as reported by Pereira et al. (New Biotechnology, 2020, 56, 79-86 doi.org/10.1016/j.nbt.2019.12.003), which is quite economical. Of course, the influence of the non-SCOAs components of the fermented stream in PHAs production should be assessed, and if any negative effect is observed further studies are required.

Reviewer comment:

5, ‘3.4.2’ to ‘3.3.2’.”

Authors answer:

The correction was made according to the reviewer’s suggestion.

Round 2

Reviewer 1 Report

Comments and Suggestions for Authors

I suggest changing 11337 (g-force for centrifugation) to 11300

Reviewer 2 Report

Comments and Suggestions for Authors

I am ok with the revision and the article can be accepted for possible publication.